# The Use of Unique, Environmental Lactic Acid Bacteria Strains in the Traditional Production of Organic Cheeses from Unpasteurized Cow’s Milk

**DOI:** 10.3390/molecules27031097

**Published:** 2022-02-07

**Authors:** Anna Łepecka, Anna Okoń, Piotr Szymański, Dorota Zielińska, Katarzyna Kajak-Siemaszko, Danuta Jaworska, Katarzyna Neffe-Skocińska, Barbara Sionek, Monika Trząskowska, Danuta Kołożyn-Krajewska, Zbigniew J. Dolatowski

**Affiliations:** 1Department of Meat and Fat Technology, Prof. Waclaw Dabrowski Institute of Agriculture and Food, Biotechnology—State Research Institute, 02-532 Warsaw, Poland; anna.okon@ibprs.pl (A.O.); piotr.szymanski@ibprs.pl (P.S.); zbigniew.dolatowski@ibprs.pl (Z.J.D.); 2Department of Food Gastronomy and Food Hygiene, Institute of Human Nutrition Sciences, University of Life Sciences-SGGW, 02-776 Warsaw, Poland; dorota_zielinska@sggw.edu.pl (D.Z.); katarzyna_kajak_siemaszko@sggw.edu.pl (K.K.-S.); danuta_jaworska@sggw.edu.pl (D.J.); katarzyna_neffe_skocinska@sggw.edu.pl (K.N.-S.); barbara_sionek@sggw.edu.pl (B.S.); monika_trzaskowska@sggw.edu.pl (M.T.); danuta_kolozyn_krajewska@sggw.edu.pl (D.K.-K.)

**Keywords:** *Lactobacillus*, starters, probiotic, culture, dairy, whey, artisanal products, functional food

## Abstract

The aim of this study was to use local LAB cultures for the production of organic acid-rennet cheeses from unpasteurized cow’s milk. Under industrial conditions, three types of cheese were produced, i.e., traditionally with acid whey (AW), with starter culture *L. brevis* B1, or with starter culture *L. plantarum* Os2. Strains were previously isolated from traditional Polish cheeses. Chemical composition, physico-chemical, microbiological, and sensory studies during 2 months of storage were carried out. As a result of this research, it was found that the basic composition was typical for semi-hard, partially skimmed cheeses. Mainly saturated fatty acids were detected. The cheeses were rich in omega-3, -6, and -9 fatty acids and conjugated linoleic acid (CLA), and were characterized by good lipid quality indices (LQI). All of the cheeses were characterized by a high number of lactic acid bacteria, with *Enterobacteriaceae*, yeast, molds, and staphylococci contaminants, which is typical microbiota for unpasteurized milk products. Water activity, pH, and total acidity were typical. A lower oxidation-reduction potential (ORP) of cheeses with the addition of strains and stability of the products during storage were observed. The B1 and Os2 cheeses were lighter, less yellow, had a more intense milk and creamy aroma, were softer, moister, and more elastic than AW cheese. The research results indicate the possibility of using environmental LAB strains in the production of high-quality acid-rennet cheeses, but special attention should be paid to the production process due to the microbiological quality of the cheeses.

## 1. Introduction

The main goal of organic farming is to produce high-quality food using environmentally friendly farming practices. According to the assumptions of organic farming, it is unacceptable to use growth stimulants and synthetic additives in animal nutrition. Organic, low-processed products, produced in small, local production plants, the products of which are based on ancient recipes, are becoming an alternative to conventional food. The production of indigenous products is a variety and tradition that have been preserved for centuries, and at the same time constitute national treasures and cultural heritage that must be especially cared for. The marketing of organic products is mainly based on sales from farms, and sales to organic stores, plants, and marketplaces [1,2].

A specific microbiological structure has developed in individual climatic regions [3]. With the current state of knowledge and technological level, it is possible to modulate the microbiome through the right diet or by taking probiotics and prebiotics. The addition of two or three species or strains of bacteria in technological processes, especially of unknown origin, does not ensure full restoration of the environmental microbiota normally contained in organic and poorly processed food. In addition, the production of such food with an extended shelf life and containing bacteria species and strains unified throughout the country or the world, has nothing to do with preserving the environmental biodiversity of food to which our ancestors have become accustomed for generations. Such a sudden change in the microbiological composition provided with food is a stress condition for our body. Considering the latest reports of scientists on the impact of biodiversity on intestinal microbiota and human health, it seems that the solution that could help to avoid many civilization diseases would be to eat organically produced, minimally processed, pasteurized food that originates from the areas where our ancestors lived for generations [4]. Research confirms that the microbiome is flexible and can change due to environmental changes [5]. The flexibility of the microbiome is a broad phenomenon that contributes to the ability of organisms to respond and adapt to environmental changes that can occur over much shorter time scales than traditional body adaptation processes. Ziegler et al. [6], defined “microbiome flexibility” as “the potential for dynamic restructuring of the host microbiome in the face of environmental change”, and “microbiome conformer” as “host species that show microbial adaptation to their surrounding environment”. The results of scientific research illustrate the importance of natural lactic acid bacteria (LAB) isolates as a valuable source of strains with new properties, providing a deeper and more complete insight into the functioning and organization of the complex metabolic system of the microbiome [2].

The specificity of a product and production technology determine the composition of the cultures used. Many types of fermented food use the most common species of microorganisms that are the natural microbiota of the raw materials. Their spontaneous development and enzymatic activity were used unknowingly by humans in antiquity, producing curdled milk, cheese, or wine in the household. The enzymatic activity of environmental bacterial cultures, controlled by appropriately selected parameters of technological processes, causes a number of changes in food, i.e., changes in the structure, consistency, sensory characteristics, and nutritional values [3,7].

At the same time, bio conservation processes take place. Cheese makers would like to use commercial starter strains to better control the fermentation process, but commercial starters are suitable for industrial production and are not suited to traditional cheese production. Indeed, commercial starter strains can dominate the autochthonous milk microbiota, ensuring a high level of safety, but at the same time causing undesirable standardization of the final product [8]. Typically, cheese is made by curdling directly whole raw milk without any heat treatment. Considering that most cheeses are commercially produced on a large scale, it is expected that the treatment of the milk allows the manipulation of the characteristics of the milk, standardizes its quality, and reduces the microbiological risk. However, to this day, the vast majority of artisan cheeses are produced using raw milk, and the leaven is whey, called natural whey starter (NWS) [9,10,11]. Starter lactic acid bacteria (SLAB) are responsible for the conversion of lactose to lactate during milk fermentation, and they lower the pH; lactic acid are responsible for the degradation of casein to peptides and free amino acids. The population of starter microorganisms is dominant in the initial period of production and is gradually reduced by changing environmental factors in the process of its maturation. Unlike SLAB, non-starter lactic acid bacteria (NSLAB) are the most difficult-to-control factor in cheese production, they can cause cheese defects and problems with maintaining their quality standards. NSLAB are practically not involved in the acidification of milk, but in the maturation process their number increases rapidly and significantly exceeds the starter microbiota [12].

According to Mora et al. [13] and Poltronieri et al. [14], there is a strong need for new strains isolated from the wild to increase the genetic diversity of the collection of lactic acid bacteria intended for traditional and new dairy applications. Therefore, the aim of this study was to use local, environmental LAB starter cultures for the production of organic acid-rennet cheeses from unpasteurized cow’s milk. A novelty in the research was the use of unique strains of lactic acid bacteria, previously isolated from Polish regional mountain cheeses, i.e., bundz and oscypek, representing potentially probiotic and antimicrobial properties. The hypothesis is as follows: The addition of these starter cultures, firstly, will increase the health promoting properties of the tested cheese samples, secondly, enrich the cheese samples with local, beneficial bacteria strains, and therefore, will serve as a vehicle to rebuild the human microbiome.

## 2. Results and Discussion

### 2.1. Chemical Composition

Table 1 shows the content of water, protein fat, cholesterol, NaCl, phosphorus, and lactose in the tested cheeses. The examined cheeses were semi-hard cheeses, partially skimmed [15]. The water content in all of the samples was in the range of 42.98–44.68%, moreover, the Os2 cheese differed significantly in water content (44.68%) from the AW and B1 cheeses (*p* < 0.05). The protein content (26.41–26.93%) in the cheeses was similar, while the B1 cheese had the most significant and highest fat content (23.28%) as compared with the AW and Os2 cheeses (21.65 and 21.25%, respectively). The examined cheeses had a fairly low cholesterol content, and the lowest cholesterol content was in the Os2 cheese (47.05 mg/100 g of product). The lowest NaCl and phosphorus contents were found in the B1 cheese (1.15 and 1.19%, respectively). No lactose was found in the tested cheeses (<0.5 mg/100 g of product) or this value was below the detection level. The chemical composition of the produced cheeses was typical for traditional, regional semi-hard cheeses. The products were high in protein, whereas they were fairly low in salt, phosphorus, and cholesterol. Interestingly, no lactose was found after the production, which proves the ability of LAB strains to completely ferment lactose contained in milk (lactose content in cow’s milk is 4.6–4.9% [16]). The resulting lactic acid significantly lowered the milk’s pH and acidity, which was important during the preservation of the product and in shaping the sensory quality.

### 2.2. The Fatty Acid Composition

The complete fatty acid profile is presented in the Appendix A. An analysis of the fatty acid profile found that the main fatty acids present in the cheese samples were saturated fatty acids (SFA), which accounted for 58.40 to 59.95% (Table 2). A statistically significant increase in the amount of SFA after storage (*p* < 0.05) was found. The Os2 cheese had the lowest SFA content (*p* < 0.05). At the same time, decreased (*p* < 0.05) monounsaturated (MUFA; 26.70–27.48%) and polyunsaturated (PUFA; 3.68–4.48%) fatty acids during storage were observed. The content of the trans fatty acids ranged from 6.73 to 7.10%, while the B1 or Os2 treatment had a significantly lower content of trans fatty acids (*p* < 0.05). The tested cheeses contained omega-3 (1.40–1.60%), omega-6 (2.15–2.85%), and omega-9 (20.35–21.18%) fatty acids, which are beneficial for human health, but their content was significantly reduced during storage of the cheeses (*p* < 0.05). The tested cheeses showed a high content of conjugated linoleic acid ((CLA) C18:2 c9t11), ranging from 2.60 to 2.85% (Appendix A). The Os2 cheese had the highest content of CLA, amounting to 2.85% after 1 month and 2.80% after 2 months of storage, respectively (*p* < 0.05).

Fatty acids play a positive or negative role in preventing and treating disease. Some fatty acid metabolites have anti-inflammatory and neuroprotective effects [17]. The main fatty acids in the tested cheeses were SFA, which is also typical for dairy products. At the same time, a high content of MUFA and PUFA was found, with a low content of trans fatty acids. In general, cheeses are associated with a high concentration of long chain fatty acids. Moreover, the composition of fatty acids depends on the animal breed, season, animal diet and species, and production processes [18].

Conjugated linoleic acid (CLA), a milk-derived fatty acid with health-promoting properties, has also been found [19]. During maturation, CLA can be formed from linoleic acid through the action of primary or additional bacterial cultures. Then, the lipid profile changes as a result of the lipolysis reaction. It was found that some *Lactobacillus* bacteria have the ability to synthesize it, however the mechanisms of this bioconversion have not been elucidated for each LAB species [20]. CLA is produced by bacteria in the rumen and is transferred to meat and dairy products from ruminants. The daily intake of CLA from the diet is unknown, because the presence of isomers in food and the factors influencing their levels are not well understood. Nevertheless, the concentration of CLA, judged to be beneficial to health by some, is around 3–6 g/day [21]. The high content of omega-3, -6, and -9 fatty acids also contributes to the favorable profiles of fatty acids. The milk fat, and especially the fatty acid profile, are essential for the sensory properties and the development of the correct flavor during maturation. Fat is not only a source of flavors, but also a solvent for fat-soluble flavor-forming substances [22].

### 2.3. Lipid Quality Indices

The tested cheeses were characterized by a fairly low AI, ranging from 2.10 to 2.21, depending on the type of cheese and storage time, but these differences were not statistically significant among the treatments (*p* > 0.05) (Table 3). The B1 cheese was characterized by lower TI (*p* < 0.05), both after storage and between the types of cheese. The cheese samples were characterized by a high DFA (41.43–42.95) and a low OFA (36.75–37.75), as well as a high H/H ratio (0.62–0.67). The HPI ranged from 0.45 to 0.48 and no significant differences were observed between the treatments and after storage (*p* > 0.05). A good indicator for dairy products is the HPI, which according to research by Chen and Liu [17], ranges from 0.16 to 0.68, depending on the type of product. The higher the HPI index, the better the impact on human health. Our tested cheeses had an HPI in the range of 0.45–0.49, with no significant differences due to product storage. The PUFA/SFA ratio is typically used to assess the effects of diet on cardiovascular health. It has been hypothesized that all PUFAs in the diet can lower low-density lipoprotein cholesterol and lower serum cholesterol, while all SFAs contribute to high serum cholesterol. Thus, the higher this index is, the more positive are the health effects [17]. The tested cheeses were characterized by good quality indicators (milk and creamy odor, softness, moisture, and elasticity), which remained at a similar level during storage, and therefore, proves the storage stability of the products. Similar research results were obtained by Mileriene et al. [23] who produced cheese. The AI index ranged from 1.93 to 2.77, while the TI index ranged from 1.58 to 2.52. In a Polish study, Paszczyk et al. [24] analyzed the lipid quality indices in smoked and non-smoked cheeses and in smoked and non-smoked cheese-like products. The AI index ranged from 0.98 to 3.18, while the TI index ranged from 2.00 to 3.86 and the Hypocholesterolemic/hypercholesterolemic ratio (H/H) ranged from 0.41 to 1.00.

### 2.4. Microbiological Analyses

The examined cheeses were characterized by a high number of microorganisms, in particular lactic acid bacteria (~8 log CFU g^−1^) as well as bacteria from the *Enterobacteriaceae* family (4.42–5.70 log CFU g^−1^), yeast and molds (3.79–4.85 log CFU g^−1^), and coagulase-positive staphylococci (3.42–5.32 log CFU g^−1^) (Table 4). Molds and acidic bacteria are ubiquitous and could certainly develop on the surface of the cheese, especially as it is moist and ripens in moderate coolness [25].

Cheeses with the addition of the B1 and Os2 strain were characterized by a higher number of LAB during storage, but the results were not statistically significant (*p* > 0.05). The addition of B1 and Os2 culture starters increased the LAB abundance. It is known that the milk used for the research was unpasteurized, so microorganisms present in the raw material may also develop. Therefore, in the control sample with whey (AW treatment), the number of LABs was also high. The initial number of bacteria contained in the starter culture guarantees that the process would be carried out by bacteria present in the starter, which guarantees the repeatability and stability of the product. The quality of the cheeses also showed the differences between the treatments, which were probably due to the addition of the different starter cultures. According to Salazar et al. [26], the addition of a known starter culture to unpasteurized milk makes it dominant in the developing product. In addition, Vázquez-Velázquez et al. [27] found that the use of a starter culture helped keep the total pathogenic microbes in pasteurized cheese below standard maximum values. At the same time, a high number of lactic acid bacteria was found. The sensory judges did not find any differences between the cheese made of unpasteurized milk and that made of pasteurized milk with the addition of a starter culture.

According to Kilcawley [28], cheeses made from raw milk are richer in microbiota, mature faster, and develop a more intense flavor and more aromatic compounds than cheeses made of pasteurized or microfiltered milk. Dairy products, especially those made from raw milk, are abundant in beneficial microorganisms (e.g., non-starter lactic acid bacteria) and in undesirable microbiota. Lactic acid bacteria are naturally present in milk, but their origin, milk quality, as well as environmental (animal diet, season, and pasture height) and hygiene conditions have a significant influence on the milk microbiota [10].

A qualitative microbiological analysis did not reveal the presence of *Salmonella* spp. and *L. monocytogenes*. Psychrotrophic bacteria have an impact on the shelf life of many food products, including dairy ones, as they can multiply at refrigerated temperatures (4–10 °C). In order to maintain the high microbiological quality of products, strict control of the raw material and the course of the technological process should be carried out, and particular emphasis should be placed on the preservation of the cold chain [29]. The cheeses were made from raw, unpasteurized cow’s milk. Most of all, acidifying microbiota is associated with milk. The use of unpasteurized raw material, on the one hand, carries a wealth of microbiota, which is so important for the sensory quality of the product, but on the other hand, carries the risk of inadequate microbiological quality [30], which is reflected in the results of our research. As the examined cheeses were not pasteurized or preserved in any way, there was a risk of primary contamination (from the raw material) or secondary contamination (e.g., during production or packaging). 

In all of the examined cheese samples, a fairly high number of coagulase-positive staphylococci (3.42–5.32 log CFU g^−1^) was found. *S. aureus* is ubiquitous in the environment and can be found in water, air, humans, and animals. It is also one of the leading causes of mastitis in cattle, and therefore, raw milk and raw dairy products can be contaminated with *S. aureus.* This contamination is most common among unpasteurized dairy products, which is especially true of products manufactured largely by hand [31]. Moreover, it has been observed that the number of staphylococci increases during the storage of products [32] which was also observed in our research. This is due to the ability of *S. aureus* to grow rapidly in milk and dairy products, and also due to the contamination of cheese makers’ skin. The enterotoxins which can be produced by staphylococci under appropriate conditions, are particularly dangerous to human health. *S. aureus* is capable of producing enterotoxins in the temperature range of 10–46 °C (optimum 34–40 °C), pH below 5 and above 9.6, and a_w_ not less than 0.86, and with the NaCl concentration above 12% [33]. By analyzing the given optimal conditions for the development of enterotoxins, it can be presumed that the cheeses produced in the present study should be free of staphylococcal toxins. However, to avoid any potent risk to consumer health we decided not to subject the produced cheeses to a sensory evaluation in terms of taste. The next stage of industrial research will be an attempt to eliminate staphylococci from unpasteurized milk, which will enable a full sensory evaluation of the products.

### 2.5. Physico-Chemical Analyses

Table 5 shows the values of water activity (a_w_), pH, acidity, and the oxidation-reduction potential (ORP). The produced cheeses were characterized by high water activity, from the start of the production process and throughout the storage period (0.93–0.96); however, the high water activity did not guarantee microbiological stability of the product [25]. The AW treatment had the highest water activity (0.94–0.96), but the cheeses with the addition of strains (treatment B1 and Os2) did not differ significantly (*p* > 0.05). During storage, a decrease in a_w_, in all cheeses, was noted, but these differences were not statistically significant (*p* > 0.05). The rapid reduction in a_w_ can be explained by water loss by evaporation and the hydrolysis of proteins to peptides and amino acids and triglycerides to glycerol and fatty acids [34]. The high a_w_ of cheeses is characteristic of traditional acid-rennet cheeses, and therefore, requires additional protection, where the preservative agent may be lactic acid bacteria in high numbers, as was the case in the present cheese production. The cheese produced on the basis of whey (AW treatment) was characterized by the lowest pH, both immediately after production and after storage (*p* < 0.05). In all of the samples, a significant increase in pH was observed after 1 month of storage, and then, a decrease in pH after 2 months of storage to the value of 5.13–5.34.

Similar results were obtained in the acidity test. In the case of the AW and B1 cheeses, an increase in acidity was observed after 1 month of storage (from 103.50 and 86.00 °SH to 106.25 and 93.50 °SH, respectively), and then, a decrease in acidity after 2 months of storage (99.00 and 86.75 °SH, respectively). The acidity of the Os2 cheese decreased significantly after 1 month of storage (to 72.25 °SH, *p* < 0.05), and after 2 months of storage it was 79.75 °SH. The pH values of the cheeses after production and during storage were above 5, which is the typical pH value for unpasteurized, semi-hard cow cheeses. Lowering the pH of milk through the production of lactic acid directly affects the stability of casein micelles and milk minerals [35]. The variation in pH value depends on the buffering capacity of the cheese and the amount of protein. The acidification process, which is continued during the maturation process, is associated, apart from high lactic acid production, also with low cheese mass buffering capacity. This is a consequence of the demineralization that occurs during the coagulation and removal of the whey. The acidity of cheeses changes during storage and, similar to the pH values, it may rise at first and then fall [35].

In all of the cheese samples, the oxidation-reduction potential increased significantly (*p* < 0.05) during storage (Table 5). Initially, the highest ORP value was characteristic for the AW treatment (338.03 mV), and the lowest Os2 cheese (258.80 mV, *p* < 0.05). After 1 month of storage, a significant increase in ORP (*p* < 0.05) was observed, especially in the case of the B1 and Os2 treatments (456.93 and 428.50 mV, respectively), followed by a slight decrease after 2 months of storage. The oxidation-reduction potential during storage increased significantly (*p* < 0.05). This is due to the microbial and enzymatic activity of a very high number of microorganisms, in particular lactic acid bacteria [36,37,38]. The high fat content of cheese may also be a predisposing factor to lipid oxidation [39]. In the tested cheeses, no additional substance was used to inhibit the oxidation processes, and the high values of the redox potential proved their advantage over the reduction processes. This increase was lower in the samples of cow cheeses with the addition of bacterial cultures (B1 and Os2 treatment) as compared with the AW treatment. As the standardization of cheese production of this type is difficult to perform [30], one of the solutions is the use of environmental lactic acid bacteria as starter cultures.

### 2.6. Measurement of Colour, Instrumental Texture Evaluation, and Sensory Evaluation

As a part of the research, it was found that the tested cheeses darkened significantly during storage (*p* < 0.05). The difference in the brightness of the cheeses was 7.66 for the AW cheese, 10.17 for the B1 cheese, and 10.83 for the Os2 cheese, respectively (time 0 and 2 were compared). Cheeses made with the strain addition (B1 and Os2) significantly differed in brightness immediately after production (L* = 78.94 and 78.33, respectively, *p* < 0.05) as compared with the whey cheese (AW, L* = 74.89), while after 2 months of storage no significant differences were noted in the brightness of the cheeses (*p* > 0.05) (Table 6). In terms of the a* parameter, it was found that the samples differed significantly from each other immediately after production. The color of the cheeses was greener (a* = −0.44 − 0.18) and, as the cheeses were stored, the a* parameter decreased significantly (values ranged from −0.51 to −0.57, *p* < 0.05). Regarding the b* parameter, the yellow color changed during storage, i.e., after 2 months of storage, the cheeses were significantly less yellow than the fresh cheeses (b* = 19.33-20.53, *p* < 0.05). Immediately after production, the most yellow cheese was the AW cheese (b* = 20.78), while after storage, the most yellow cheese was the B1 cheese (b* = 20.69).

The texture parameters of the tested acid-rennet cheeses were similar and increased during the storage of the products (Table 6), with the exception of the adhesiveness of the products, which after production had the highest value for the Os2 cheese (1.58 mJ) and the lowest after storage (0.58 mJ), unlike other products. Among the tested products, the AW cheese was characterized by the highest value of hardness 1, elasticity, and chewiness (65.08 N, 8.57 mm, and 332.08 mJ, respectively). In addition, the value of the cohesiveness of the Os2 and B1 cheeses was similar and remained almost at the same level during the storage of the products, unlike the AW cheese, in which the cohesiveness value decreased during storage (0.44). The color and texture are important cheese quality distinguishing features, as they are important for consumers when deciding to buy a product. The color itself is one of the first quality attributes on the basis of which the consumer assesses the acceptability of a product [7]. According to Pérez-Soto et al. [40], the addition of bioactive compounds may affect the hardness of cheeses, but it has no effect on elasticity, firmness, and chewiness. However, it depends on the type of cheese. The color of the cheeses can be related to various internal and external factors, and the opacity of the cheeses (L*) can be affected by the degree of matrix hydration. The differences may be mainly due to the initial composition of milk and whey and the technology of cheese making [41]. The b* color parameter is strongly correlated with the yellow color and may be related to maturation time. Springiness is a measure of the return to its original, undeformed state after the first compressive force is removed, and cohesiveness is the amount by which the cheese can be deformed before breaking. These characteristics are influenced by the composition of the milk, cheese production and procedures, microorganisms, maturation and moisture, pH, and soluble calcium. Flexibility shows whether the biochemical reactions taking place inside the cheese are not sufficient to modify the final structure, giving more or less flexibility [34]. The texture parameters changed during the storage of the products, which correlated with the results of other physicochemical analyses. From a consumer’s point of view, a cheese that is lighter and less yellow is preferable. The tested cheeses with the addition of the B1 and Os2 strains had more consumer preferable parameters than the cheese with whey.

Figure 1 shows the results of the sensory evaluation. The color of the cheeses was towards the yellow (7.54–7.84 c.u.). After storage, a significantly more yellow color of the cheeses was found (8.12–8.32 c.u., *p* < 0.05). The cheeses produced with the addition of the B1 or Os2 strains had a significantly more intense milk and creamy odor. After storage, a significantly more intense milk odor was found (6.87–7.69 c.u., *p* < 0.05). The cheeses produced with the addition of the B1 and Os2 strains were more moist, elastic, and softer. The performed sensory analysis of the products does not give a complete picture of the sensory profile, due to the presence of *S. aureus*; a taste assessment of the products was not carried out.

Acid-rennet cheeses are products whose appearance, physical, and organoleptic characteristics depend largely on the multidirectional enzymatic and metabolic activity of cultures and enzyme preparations used in their production [42]. During maturation, the texture, structure, taste, and aroma of the cheeses changed. Changes in the structure and consistency of cheeses depend on both technological operations and the nature of fermentation processes, CO_2_ formation, and the degree of protein degradation. Organoleptic characteristics are shaped by numerous end metabolites of lactic acid fermentation, proteolysis, lipolysis, and sometimes propionic fermentation. The correct proportions of compounds formed in these transformations determine the characteristic taste and smell of various types of cheese.

## 3. Materials and Methods

### 3.1. Materials

#### 3.1.1. Acid Whey

The acid whey was obtained from the previous production of acid-rennet cheeses. The average pH value was 4.79 ± 0.12. The total number of microorganisms was 8.71 ± 0.62 log CFU mL^−1^, the number of lactic acid bacteria was 8.14 ± 0.95 log CFU mL^−1^, and the number of yeasts and molds was 5.46 ± 0.77 log CFU mL^−1^. The whey did not contain the pathogenic microorganisms *L. monocytogenes*, *Salmonella* spp., and *S. aureus*. The method of pH measurement and the method of microbiological analysis are presented below in the Methodology section (Section 3.2.4, “Microbiological analyses”).

#### 3.1.2. Starters Lactic Acid Bacteria Strains Preparation

Two environmental LAB strains were selected for the production of acid-rennet cheeses: *Levilactobacillus brevis* B1 and *Lactiplantibacillus plantarum* Os2. The strains came from the internal collection of microorganisms of the Department of Food Gastronomy and Food Hygiene, Institute of Human Nutrition Sciences, WULS-SGGW. *L. brevis* B1 was isolated from ”bundz” (Polish, regional cheese) and was characterized by a high capacity for milk fermentation. In the study by Zielińska et al. [43], the impact of fructose and oligofructose addition on the physicochemical, rheological, microbiological, and sensory properties of fermented milk products inoculated with own probiotic *Lactobacillus* starter cultures was assessed. The highest overall levels of sensory quality were observed for *L. brevis* B1 samples supplemented with oligofructose. *L. plantarum* Os2 was isolated from ”oscypek” (a Polish, smoked cheese made of salted sheep milk exclusively in the Tatra Mountains region of Poland) and was characterized by high proteolytic and saccharolytic activity, and exhibited antimicrobial properties, especially against the *L. monocytogenes* [44]. In a study by Ołdak et al. [45], they showed that *L. plantarum* strains or their metabolites could potentially be used in the food industry as protective cultures to extend the shelf-life of foodstuffs, in particular fermented dairy products. The *L. plantarum* Os2 strain showed strong anti-staphylococcal properties.

The strains were stored at −80 °C with a 20% addition of glycerol (Sigma-Aldrich, Saint Louis, USA). MRS broth (LabM, Heywood, UK) was used to cultivate the cultures and incubated at 37 °C for 24 h. Then, bacterial cells were washed three times in PBS (phosphate buffered saline, pH 7.4, Sigma-Aldrich, Saint Louis, USA) and diluted in milk to a final concentration of 8 log CFU mL^−1^.

#### 3.1.3. Production of Acid-Rennet Cheeses from Cow’s Milk

As part of the research, three types of cheeses were made. The first (AW treatment) was inoculated with acid whey, while the second and third were acidified with the addition of *L. brevis* B1 (B1 treatment) or *L. plantarum* Os2 (Os2 treatment). The cheeses were produced in a family-run organic farm, located in the south of Poland. The farm specializes in the traditional processing of unpasteurized milk. The farm produces white cheeses, rennet cheeses (bundz and bryndza), ripening cheeses, cottage cheese, homemade butter, raw milk, and whey.

Acid whey (~8 log CFU mL^−1^) or LAB starter sourdough (~8 log CFU mL^−1^) was added to raw, unpasteurized cow’s milk and left at room temperature for 16–18 h. Then, the milk was heated to 40 °C and 0.625 g of microbial rennet was added. The mixture was thoroughly mixed and left for 20–40 min in order to form a clot. The curd was mixed with a ferula. After 5 min, 100 mL of boiling water was poured on the surface, causing the curd to settle. The precipitated curd was put into molds made of stainless steel, and covers were put on and secured. In order to seal the cheese, the molds were poured over with boiling water and then pressed for 4–6 h. After the cheeses were pressed, they were stored in brine (saturated rock salt solution) for 6–8 h. Then, the cheeses were matured at a temperature of 8–10 °C and were tested immediately after production and after 1 and 2 months of storage. 

### 3.2. Methodology

#### 3.2.1. Chemical Composition

The water content was determined by the weighing method according to ISO 1442:1997 [46]. The method consisted of thoroughly mixing the sample with sand and drying it to a constant mass in a laboratory dryer at 103 °C. The value is presented in %.

The protein content was determined by the Kjeldahl method in accordance with PN-A-04018:1975/Az3:2002 [47]. The method consisted in determining the total nitrogen content, and then, using a factor of 6.25 to calculate the protein. The value is presented in %.

The content of free fat was determined by the Weibull–Berntrop gravimetric method according to ISO 8262-3:2005 [48]. The value is presented in %.

The cholesterol content was determined by extracting the lipid fraction from the sample, esterifying fatty acids, and derivatizing cholesterol in the presence of an internal standard. The sample was analyzed by gas chromatography with flame ionization detection (GC-FID). The cholesterol value is expressed in mg/100 g of product.

The chloride content was determined using the potentiometric method according to ISO 1841-2:1996 [49]. The value is presented in %.

The phosphorus content was determined in accordance with PN-A-82060:1999 [50]. The sample was burned in a combustion furnace at a temperature of 525 °C. After cooling, nitric acid was added to the ashed sample, and it was heated in a boiling bath. Then, the crucible contents were filtered through filter paper. A blank solution was prepared in parallel. A precipitating reagent was added to the resulting solutions and placed on a heating plate and boiled. The samples were cooled to room temperature. By means of a glass rod, a pellet was transferred to a Gooch funnel, previously dried to a constant weight at 250 °C. The funnel with the sediment was dried at a temperature of 250 °C until constant weight, then, after cooling in a desiccator, it was weighed on an analytical balance. The content of total phosphorus converted to P_2_O_5_ was calculated according to the formula:X = 2.29 (1.4 × (A − B)/M) [%](1)
where A is the mass of dried sludge from the test sample; B is the mass of dried sludge from the blank sample; M is the mass of the sample; 1.4 is the coefficient, being the product of the gravimetric coefficient and conversion into percentages; 2.29 is the conversion factor for P_2_O_5_.

The lactose content was determined by liquid chromatography using the RID refractive index detector. Lactose was separated by column chromatography and determined by an external standard method. The lactose value is expressed in mg/100 g.

#### 3.2.2. The Fatty Acid

Composition was determined by the GC method (HP 6890 II with a flame-ionization detector, Hewlett-Packard, USA) according to ISO 12966-1:2014 [51]. A BPX70 high-polar capillary column (60 m × 0.25 mm, 25 μm) was used to separate the esters. The injector temperature was 240 °C with a split ratio set to 100:1, FID temperature was 250 °C. The oven temperature was increased from 130 °C (1 min) to 210 °C (7 min) at a rate of 1.5 °C min^−1^. As a carrier, gas helium was used with a constant pressure of 40 psi at a flow rate of 0.3 mL min^−1^ and an injection volume of 1 μL. GC-FID was used to calculate the relative percent areas of the FAME components. The analysis time was 61 min. The results were calculated according to the principle of internal normalization by the ChemStation software (A 03.34^®^ 1989–1994). Peaks of fatty acid methyl esters resolved by GC were identified by comparing to standard FAME mixtures of known composition. The manuscript summarizes the results (sum of saturated, monounsaturated, polyunsaturated fatty acids, sum of trans, omega-3, omega-6, and omega-9 fatty acids). The complete fatty acid profile is presented in the Appendix A. The values are presented in %.

#### 3.2.3. Lipid Quality Indices

These were calculated from the following formulas [17,52,53]:

The index of atherogenicity
AI = (C12:0 + (4 × C14:0) + C16:0)/(Σ n − 3 PUFA + Σ n − 6 PUFA + Σ MUFA)(2)

The index of thrombogenicity
TI = (C14:0 + C16:0 + C18:0)/[(0.5 × C18:1) + (0.5 × other MUFA) + (0.5 × Σ n − 6 PUFA) + (3 × Σ n − 3 PUFA) + Σ n − 3 PUFA/Σ n-6 PUFA)](3)

Hypocholesterolemic fatty acids
DFA = UFA + C18:0(4)

Hypercholesterolemic fatty acids
OFA = C12:0 + C14:0 + C16:0(5)

The ratio of hypocholesterolemic and hypercholesterolemic fatty acids
H/H = (C18:1 n − 9 + C18:2 n − 6 + C18:3 n − 3)/(C12:0 + C14:0 + C16:0)(6)

Health-promoting index
HPI = Σ UFA/[C12:0 + (4 × C14:0) + C16:0](7)

#### 3.2.4. Microbiological Analyses

The plate culture technique was used to determine the number of microorganisms. For the enumeration of *Enterobacteriaceae* (ENT), MacConkey agar No. 3 (LabM, Heywood, UK) was used, in accordance with ISO 21528–2:2017 [54]; for the determination of the number of yeasts and molds (TYMC), YGC agar (Sabouraud Dextrose with Chloramphenicol LAB-Agar, Biomaxima, Lublin, Poland) was used, in accordance with ISO 21527–1:2008 and ISO 21527-2:2008 [55,56]; for the enumeration of coagulase-positive staphylococci (*Staphylococcus aureus* and other species), in accordance with ISO 6888–1:2021 [57], Baird–Parker agar medium with 5% Egg Yolk Tellurite (LabM, Heywood, UK) were used. The number of microorganisms is expressed as log colony forming units per gram (log CFU g^−1^).

The presence of pathogenic bacteria was determined by the method of enrichment of the culture on the media indicated in the standards. XLD agar (Xylose Lysine Deoxycholate Agar, LabM, Heywood, UK) and RAPID’*Salmonella* agar (Bio-Rad, Hercules, CA, USA) were used to determine the presence of *Salmonella*, according to ISO 6579-1:2017 [58]; ALOA agar (*Listeria* according to Ottaviani and Agosti Agar, Bio-Rad, Hercules, CA, USA) and PALCAM agar (LabM, Heywood, UK) were used to determine the presence of *Listeria monocytogenes* according to ISO 11290-2:2017 [59].

#### 3.2.5. Physico-Chemical Analyses

For water activity, an AQUALAB Pawkit water activity meter (METER Group, Inc., Washington, DC, USA), according to the ISO 18787:2017 standard [60], was used to determine the water activity (a_w_) of the cheese samples. Samples were taken immediately after opening the package and placed in special, closed vials. Then, the samples were kept at 25 °C for 1 h to obtain the appropriate temperature.

For pH values, a sample of 30 g of cheese was thoroughly ground with a pestle, gradually adding in small amounts 30 cm^3^ of distilled water at 40 °C until a homogeneous emulsion was obtained. The resulting emulsion was brought to a temperature of 20 °C and measured with a SevenCompactTM S220 with an InLab electrode (Mettler Toledo, Ohio, USA).

For titratable acidity, a sample weighing 5 g was ground with a pestle, adding 50 cm^3^ of distilled water at 40 °C in small amounts until a homogeneous emulsion was obtained. Then, 2 cm^3^ of a 2% alcoholic phenolphthalein solution was added and titrated with 0.25 M NaOH solution until the color remained slightly pink for 30 s. The values are given in °SH.

For the oxidation-reduction potential (ORP), a 30 g sample was ground with a pestle, gradually adding in small amounts 30 cm^3^ of distilled water at 40 °C until a homogeneous emulsion was obtained. The resulting emulsion was brought to a temperature of 20 °C and measured with a SevenCompact ™ S220 (Mettler Toledo, OH, USA) with an InLab Redox Pro electrode. The ORP value is expressed in mV.

#### 3.2.6. Instrumental Measurement of Color

To measure the color components in the CIELab system, L* (lightness), a* (chromaticity from green (−a) to red (+a)), and b* (chromaticity from blue (−b) to yellow (+b)) were performed using a Minolta CR-300 spectrophotometer (Konica Minolta, Tokyo, Japan). During the measurements, a standard CIE observer was used: 2 °, illuminant D65, and diaphragm diameter 8 mm. The calibration was done with white tile standard (L* 99.18; a*, 0.07; b*, 0.05).

#### 3.2.7. Instrumental Texture Evaluation

A texture profile analysis (TPA) was performed with a CT3 Texture Analyzer (Brookfield Ametek, USA). The samples were compressed at a speed of 0.50 mm s-^1^ twice to 50% of their original height with a cylindrical head 38.1 mm in diameter and 20 mm high at a maximum pressing force of 10,000 g. The tested cheese was shaped into a 20 mm square. The following texture parameters were determined: hardness 1 (force required to obtain deformation (N)); hardness 2 (force necessary to obtain deformation (N)); adhesiveness (the work necessary to overcome the attractive forces between the surface of the food and the surface of other materials with which the food comes into contact (mJ)); cohesiveness (the extent to which the material can be deformed before fracture (dimensional)); resilience (speed at which the deformed material returns to an undistorted state after the deforming force is removed (mm)); gumminess (energy needed to break up a semi-solid food product that is ready-to-swallow, i.e., a product with a low degree of hardness and high degree of cohesiveness (N)); chewiness (energy required to chew solid food until it is ready to swallow, i.e., a product of hardness, cohesiveness, and elasticity (mJ)); and average peak load (N) [61]. Measurements were carried out at room temperature 20 min after the samples were removed from the cooling chamber.

#### 3.2.8. Sensory Evaluation

The sensory quality of goat’s acid-rennet cheese was determined by incorporating the scaling method [62]. A linear scale (100 mm) converted to numerical values (0–10 c.u.) was used. The trained panel consisted of 16 members who were thoroughly and formally tested prior to their selection in accordance with ISO 8586:2012 [63]. Due to the results of the microbiological analyses, only the appearance and the smell were assessed (no taste assessment). To compare the sensory quality of the cheeses, the following parameters were used: five odor discriminants (milk, sour, creamy, irritating, and other, where 0 is not intensive and 10 is very intensive); color discriminant (where 0 is white and 10 is yellow); softness discriminant (where 0 is soft and 10 is hard); moisture discriminant (where 0 is dry and 10 is moist); elasticity discriminant (where 0 is not very flexible and 10 is very flexible). The cheese samples were divided into portions (average piece size approximately 15 g) and placed in disposable boxes made of clear food-safe plastic. Then, the boxes were sealed and individually coded with three-digit codes and given in random order for evaluation.

#### 3.2.9. Statistical Analysis

Four separate, independent cheese products were made (4 replicates) at different times, under industrial conditions, and a completely randomized design was used. A one-way analysis of variance (ANOVA) was applied, first between treatments (AW, B1, and Os2), and then, between the storage time (0, 1, and 2 months) of the samples for all parameters. Means and standard deviation were calculated and a probability level of *p* < 0.05 was used in testing the statistical significance of all of the experimental data. The Tukey’s post hoc test was used to determine the significance of the mean values for multiple comparisons (*p* < 0.05). The Statistica 13.1 (TIBCO Software Inc., Palo Alto, California, USA) program was used to perform the calculations.

## 4. Conclusions

Despite the widespread use of defined starters in the industry and the many strains available on the market to develop new products, there is an increasing demand for new cultures. The latest molecular tools are becoming effective in screening huge numbers of strains potentially useful in terms of technological as well as functional properties. At this point, it is extremely important to test the nutritional, technological, sensory, and safety aspects of environmental strains, which were carried out in these studies.

This research shows that environmental LAB strains have great potential for future applications in the dairy industry. Technologies for the production of acid-rennet cheese with the use of local environmental bacterial cultures are in line with the current trends in the development of fermented foods and human nutrition and should be continued, especially in organic food processing. The *L. brevis* B1 and *L. plantarum* Os2 strains that we have proposed have proven successful in the production of acid-rennet cow cheeses.

The cheeses with *L. brevis* B1 and *L. plantarum* Os2 were moister, and the *L. brevis* B1 cheese had a slightly higher fat content than the AW cheese. Lower cholesterol, less trans fatty acids, and more CLA (*L. plantarum* Os2 cheese) were observed as compared with the cheese with the addition of whey. All of the cheeses had good LQI and stability during the two-month storage period. Moreover, the *L. brevis* B1 and *L. plantarum* Os2 cheeses were characterized by a higher number of LAB, but at the same time they did not acidify the product so much (higher pH and lower titratable acidity). It should be noted that cheeses with the addition of selected strains had a lower oxidation-reduction potential after the storage of the products, which indicates a lower amount of secondary fat oxidation products and product stabilization. These cheeses were lighter, less yellow, had a greater intensity of a milky and creamy odor, and were moister, soft, and elastic. These sensory characteristics are important to consumers when evaluating food products. Overall, it can be concluded that cheeses produced with the addition of environmental strains were of better quality than cheeses with the addition of whey.

The hypotheses were both confirmed. First, the addition of LAB cultures improved the health-promoting properties of the product (Os2 cheese had a high CLA content, all cheeses had a lower trans fatty acid content and good LQI indexes) and allowed for the storage stability. Secondly, the cheeses were enriched with beneficial bacterial strains with potentially probiotic properties, which could be used in the reconstruction of the human microbiota. The cheeses with the addition of B1 or Os2 strains maintained a high number of LAB throughout the storage period (above 8 log CFU g^−1^).

This research will be continued in order to optimize the production process. It is particularly important to improve the hygienic conditions of the raw material and to eliminate undesirable microorganisms among cheese makers, which significantly affects the deterioration of the microbiological quality of the final product.

## Figures and Tables

**Figure 1 molecules-27-01097-f001:**
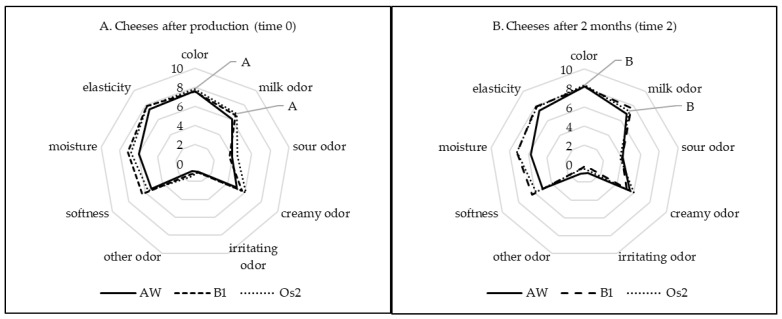
Sensory evaluation of the tested cow cheeses after production (**A**) and after 2 months of storage (**B**). AW, cheese with acid whey; B1, cheese with *L. brevis* B1; Os2, cheese with *L. plantarum* Os2. Means in the same row followed by different uppercase letters (A,B) are significantly different (*p* < 0.05), *n* = 16.

**Table 1 molecules-27-01097-t001:** The content of water, protein, fat, cholesterol, NaCl, phosphorus, and lactose in the tested cow cheeses.

Content	Cheese Symbol		
	AW	B1	Os2
Water (%)	42.98 ± 0.10 ^A^	43.75 ± 0.17 ^A^	44.68 ± 0.79 ^B^
Protein (%)	26.93 ± 0.10 ^A^	26.41 ± 0.79 ^A^	26.90 ± 0.41 ^A^
Fat (%)	21.65 ± 0.33 ^A^	23.28 ± 0.33 ^B^	21.25 ± 0.04 ^A^
Cholesterol (mg/100 g)	49.90 ± 1.89 ^B^	47.83 ± 1.66 ^A^	47.05 ± 0.68 ^A^
NaCl (%)	1.33 ± 0.05 ^B^	1.15 ± 0.06 ^A^	1.70 ± 0.00 ^C^
Phosphorus (%)	1.23 ± 0.00 ^A^	1.19 ± 0.01 ^A^	1.24 ± 0.05 ^A^
Lactose (mg/100 g)	<0.5 ^A^	<0.5 ^A^	<0.5 ^A^

AW, cheese with the addition of whey; B1, cheese with the addition of *L. brevis* B1; Os2, cheese with the addition of *L. plantarum* Os2. The values are expressed as means ±SD, means in the same row followed by different uppercase letters (^A^–^C^) represent significant differences (*p* < 0.05), *n* = 4.

**Table 2 molecules-27-01097-t002:** A summary of the fatty acid profiles in the tested cow cheeses after production and after 1 and 2 months of storage.

Sum	Cheese Symbol								
	AW			B1			Os2		
SFA	58.63 ± 0.05 ^bA^	59.15 ± 0.06 ^bB^	59.45 ± 0.17 ^aB^	59.03 ± 0.10 ^cA^	59.70 ± 0.08 ^cB^	59.95 ± 0.13 ^bB^	58.40 ± 0.08 ^aA^	58.93 ± 0.19 ^aB^	59.30 ± 0.08 ^aC^
MUFA	27.05 ± 0.24 ^aB^	27.08 ± 0.05 ^aB^	26.78 ± 0.15 ^aA^	27.18 ± 0.10 ^aB^	27.15 ± 0.13 ^aB^	26.70 ± 0.18 ^aA^	27.30 ± 0.08 ^aB^	27.48 ± 0.13 ^bC^	26.83 ± 0.10 ^aA^
PUFA	4.45 ± 0.10 ^aC^	3.98 ± 0.10 ^cB^	3.88 ± 0.05 ^bA^	4.40 ± 0.08 ^aC^	3.68 ± 0.05 ^aA^	3.80 ± 0.00 ^aB^	4.48 ± 0.10 ^aB^	3.85 ± 0.10 ^bA^	3.80 ± 0.00 ^aA^
Trans	7.08 ± 0.05 ^cA^	7.03 ± 0.05 ^cA^	7.10 ± 0.00 ^bA^	6.73 ± 0.05 ^aA^	6.73 ± 0.05 ^aA^	6.80 ± 0.08 ^aA^	6.95 ± 0.06 ^bA^	6.90 ± 0.00 ^bA^	6.88 ± 0.05 ^aA^
Ω-3	1.60 ± 0.00 ^aA^	1.50 ± 0.00 ^bA^	1.60 ± 0.00 ^aA^	1.60 ± 0.00 ^aB^	1.40 ± 0.00 ^aA^	1.60 ± 0.00 ^aB^	1.60 ± 0.00 ^aA^	1.60 ± 0.00 ^cA^	1.60 ± 0.00 ^aA^
Ω-6	2.85 ± 0.10 ^aB^	2.28 ± 0.10 ^bA^	2.28 ± 0.05 ^bA^	2.80 ± 0.08 ^aB^	2.78 ± 0.05 ^cB^	2.30 ± 0.00 ^bA^	2.88 ± 0.10 ^aB^	2.15 ± 0.10 ^aA^	2.20 ± 0.00 ^aA^
Ω-9	20.93 ± 0.13 ^aC^	20.80 ± 0.08 ^aB^	20.48 ± 0.15 ^bA^	20.93 ± 0.05 ^aC^	20.70 ± 0.14 ^aB^	20.33 ± 0.17 ^aA^	21.18 ± 0.15 ^bC^	20.85 ± 0.06 ^aB^	20.35 ± 0.06 ^aA^
	0	1	2	0	1	2	0	1	2
	Time (month)								

AW, cheese with acid whey; B1, cheese with *L.*
*brevis* B1; Os2, cheese with *L. plantarum* Os2; SFA, all saturated fatty acids; MUFA, all monounsaturated fatty acids; PUFA, all polyunsaturated fatty acids; trans, all trans fatty acids; Ω-3-6-9, all fatty acids omega-3, -6, and -9. The values are expressed as means ±SD, means in the same row followed by different uppercase letters (^A^–^C^) are significantly different (*p* < 0.05), means in the same column followed by different lowercase letters (^a^–^c^) are significantly different (*p* < 0.05), *n* = 4.

**Table 3 molecules-27-01097-t003:** Lipid quality indices in the tested cow cheeses after production and after 1 and 2 months of storage.

LQI	Cheese Symbol								
	AW			B1			Os2		
AI	2.10 ^aA^	2.14 ^aA^	2.16 ^aA^	2.13 ^aA^	2.21 ^aA^	2.13 ^aA^	2.09 ^aA^	2.09 ^aA^	2.19 ^aA^
TI	1.49 ^aA^	1.51 ^aB^	1.50 ^bB^	1.50 ^aA^	1.54 ^aB^	1.34 ^aA^	1.48 ^aA^	1.38 ^aA^	1.42 ^aB^
DFA	42.95 ^cA^	42.39 ^bB^	42.01 ^aB^	42.58 ^bA^	41.66 ^aA^	41.43 ^aA^	42.81 ^bA^	42.01 ^aA^	41.53 ^aA^
OFA	36.75 ^aA^	36.78 ^aA^	36.78 ^aA^	37.45 ^aB^	37.68 ^bB^	37.75 ^bB^	37.15 ^aB^	37.03 ^aA^	37.25 ^aB^
H/H	0.67 ^aA^	0.66 ^aA^	0.65 ^aA^	0.66 ^aA^	0.66 ^aA^	0.62 ^aA^	0.67 ^aA^	0.66 ^aA^	0.63 ^aA^
HPI	0.48 ^aA^	0.47 ^aA^	0.46 ^aA^	0.47 ^aA^	0.46 ^aA^	0.45 ^aA^	0.48 ^aA^	0.47 ^aA^	0.46 ^aA^
	0	1	2	0	1	2	0	1	2
	Time (month)								

AW, cheese with acid whey; B1, cheese with *L. brevis* B1; Os2, cheese with *L. plantarum* Os2; LQI, lipid quality indices; AI, index of atherogenicity; TI, index of thrombogenicity; DFA, hypocholesterolemic fatty acids; OFA, hypercholesterolemic fatty acids; H/H, the ratio of hypocholesterolemic and hypercholesterolemic fatty acids; HPI, health-promoting index. The values are expressed as means, means in the same row followed by different uppercase letters (^A^–^C^) are significantly different (*p* < 0.05), means in the same column followed by different lowercase letters (^a^–^c^) are significantly different (*p* < 0.05), *n* = 4.

**Table 4 molecules-27-01097-t004:** Microbiological analysis of the tested cow cheeses after production and after 1 and 2 months of storage.

Analysys	Cheese Symbol								
	AW			B1			Os2		
LAB [log CFU g^−1^]	8.64 ± 0.21 ^bA^	8.12 ± 0.19 ^aA^	7.98 ± 0.01 ^aA^	8.62 ± 0.02 ^aA^	8.70 ± 0.10 ^aB^	8.70 ± 0.02 ^aC^	8.56 ± 0.25 ^aA^	8.54 ± 0.22 ^aB^	8.48 ± 0.11 ^aB^
ENT [log CFU g^−1^]	4.89 ± 0.53 ^bA^	4.68 ± 0.45 ^bA^	4.42 ± 0.11 ^aA^	5.40 ± 0.09 ^aB^	5.70 ± 1.10 ^aB^	5.62 ± 0.17 ^aC^	5.42 ± 0.09 ^bB^	5.16 ± 0.06 ^aB^	5.00 ± 0.11 ^aB^
Y&M [log CFU g^−1^]	4.02 ± 0.17 ^aA^	4.85 ± 0.11 ^bC^	5.15 ± 0.84 ^bB^	3.88 ± 0.10 ^aA^	4.50 ± 0.10 ^bB^	3.98 ± 0.08 ^aA^	3.79 ± 0.07 ^aA^	4.04 ± 0.03 ^bA^	3.99 ± 0.26 ^bA^
STA [log CFU g^−1^]	4.12 ± 0.22 ^bA^	3.59 ± 0.16 ^aA^	3.42 ± 0.12 ^aA^	4.31 ± 0.06 ^aA^	4.93 ± 4.10 ^bB^	4.69 ± 0.20 ^aB^	4.17 ± 0.13 ^aA^	5.32 ± 0.03 ^bC^	5.05 ± 0.33 ^bC^
SALM	nd	nd	nd	nd	nd	nd	nd	nd	nd
LIST	nd	nd	nd	nd	nd	nd	nd	nd	nd
	0	1	2	0	1	2	0	1	2
	Time (month)								

LAB/ ENT/Y&M/STA, count of microorganisms; SALM/LIST, presence of *Salmonella* spp. or *L. monocytogenes* in 25 g of product; LAB, lactic acid bacteria; ENT, *Enterobacteriaceae* family; Y&M, yeast and molds; STA, coagulase-positive staphylococci; nd, not detected. The values are expressed as means ±SD, means in the same row followed by different uppercase letters (^A^–^C^) are significantly different (*p* < 0.05), means in the same column followed by different lowercase letters (^a^–^c^) are significantly different (*p* < 0.05), *n* = 4.

**Table 5 molecules-27-01097-t005:** Water activity, pH, titratable acidity, and oxidation-reduction potential of tested cow cheeses after production and after 1 and 2 months of storage.

Parameter	Cheese Symbol								
	AW			B1			Os2		
a_w_	0.96 ± 0.01 ^aA^	0.95 ± 0.01 ^aA^	0.94 ± 0.01 ^aA^	0.95 ± 0.01 ^aA^	0.95 ± 0.00 ^aA^	0.94 ± 0.01 ^aA^	0.96 ± 0.01 ^aA^	0.94 ± 0.00 ^aA^	0.93 ± 0.01 ^aA^
pH	5.03 ± 0.05 ^aA^	5.21 ± 0.02 ^aC^	5.13 ± 0.03 ^aB^	5.14 ± 0.03 ^bA^	5.45 ± 0.04^cC^	5.33 ± 0.06 ^bB^	5.08 ± 0.07 ^aA^	5.38 ± 0.02 ^bB^	5.34 ± 0.02 ^bB^
TA [°SH]	103.50 ± 16.92 ^aA^	106.25 ± 25.02 ^bA^	99.00 ± 2.16 ^bA^	86.00 ± 4.32 ^aA^	93.50 ± 4.65 ^bA^	86.75 ± 4.57 ^aA^	113.50 ± 10.63 ^bB^	72.25 ± 3.30 ^aA^	79.75 ± 8.02 ^aA^
ORP [mV]	338.03 ± 31.67 ^bA^	478.48 ± 30.29 ^bB^	453.33 ± 7.33 ^bB^	299.98 ± 8.96 ^bA^	456.93 ± 10.48 ^bC^	436.53 ± 10.43 ^bB^	258.80 ± 14.85 ^aA^	428.50 ± 4.76 ^aB^	418.53 ± 12.23 ^aB^
	0	1	2	0	1	2	0	1	2
	Time (month)								

a_w_, water activity; TA, titratable acidity; ORP, oxidation-reduction potential. The values are expressed as means ±SD, means in the same row followed by different uppercase letters (^A^–^C^) are significantly different (*p* < 0.05), means in the same column followed by different lowercase letters (^a^–^c^) are significantly different (*p* < 0.05), *n* = 4.

**Table 6 molecules-27-01097-t006:** Color and texture of the tested cow cheeses after production and after 1 and/or 2 months of storage.

Parameter	Cheese Symbol								
	AW			B1			Os2		
L*	74.89 ± 0.48 ^aC^	71.49 ± 0.15 ^bB^	67.23 ± 0.73 ^aA^	78.94 ± 0.22 ^bC^	72.86 ± 1.70 ^cB^	68.77 ± 0.40 ^aA^	78.33 ± 1.76 ^bC^	69.85 ± 0.35 ^aB^	67.50 ± 0.71 ^aA^
a*	−0.44 ± 0.03 ^aB^	−0.56 ± 0.03 ^bA^	−0.57 ± 0.02 ^aA^	0.18 ± 0.03^cC^	−0.47 ± 0.04 ^cB^	−0.57 ± 0.01 ^aA^	−0.30 ± 0.14 ^bC^	−0.79 ± 0.05 ^aA^	−0.51 ± 0.09 ^aB^
b*	20.78 ± 0.30 ^bB^	20.52 ± 0.24 ^bB^	20.02 ± 0.73 ^bA^	20.69 ± 0.13 ^bA^	20.46 ± 0.24 ^bA^	20.53 ± 0.48 ^bA^	19.98 ± 0.62 ^aB^	19.80 ± 0.50 ^aB^	19.33 ± 0.50 ^aA^
Hardness Cycle 1 [N]	65.08 ± 14.48 ^aA^	x	91.15 ± 10.82 ^bB^	60.56 ± 12.98 ^aA^	x	72.99 ± 12.43 ^aA^	57.21 ± 8.39 ^aA^	x	81.64 ± 13.08 ^aB^
Adhesiveness [mJ]	1.18 ± 0.97 ^aA^	x	1.77 ± 1.02 ^bA^	0.93 ± 0.83 ^aA^	x	1.32 ± 0.61 ^bA^	1.58 ± 0.75 ^aB^	x	0.50 ± 0.18 ^aA^
Hardness Cycle 2 [N]	48.14 ± 14.16 ^aA^	x	59.21 ± 9.44 ^aA^	48.95 ± 9.52 ^aA^	x	61.81 ± 10.00 ^aB^	45.38 ± 6.12 ^aA^	x	68.80 ± 9.78 ^aB^
Cohesiveness	0.58 ± 0.09 ^aB^	x	0.44 ± 0.05 ^aA^	0.67 ± 0.03 ^bA^	x	0.68 ± 0.02 ^bA^	0.67 ± 0.04 ^bA^	x	0.70 ± 0.02 ^bA^
Springiness [mm]	8.57 ± 0.50 ^aA^	x	13.32 ± 9.07 ^aA^	8.51 ± 0.11 ^aA^	x	10.48 ± 5.47 ^aA^	8.55 ± 0.31 ^aA^	x	8.56 ± 0.17 ^aA^
Gumminess [N]	38.45 ± 13.53 ^aA^	x	40.74 ± 8.95 ^aA^	40.47 ± 7.78 ^aA^	x	49.86 ± 8.14 ^aA^	38.20 ± 5.01 ^aA^	x	57.37 ± 8.44 ^bB^
Chewiness [mJ]	332.08 ± 121.71 ^aA^	x	594.70 ± 529.40 ^aB^	344.42 ± 67.84 ^aA^	x	537.60 ± 340.91 ^aB^	326.82 ± 45.24 ^aA^	x	491.90 ± 78.48 ^aB^
Average Peak Load [N]	52.37 ± 14.27 ^aA^	x	67.19 ± 12.31 ^aB^	51.85 ± 10.37 ^aA^	x	64.60 ± 10.54 ^aA^	46.66 ± 7.04 ^aA^	x	72.01 ± 10.68 ^aB^
	0	1	2	0	1	2	0	1	2
	Time (month)								

AW, cheese with acid whey; B1, cheese with *L. brevis* B1; Os2, cheese with *L. plantarum* Os2; L*/a*/b*, color parameters. The values are expressed as means ±SD, means in the same row followed by different uppercase letters (^A^–^C^) are significantly different (*p* < 0.05), means in the same column followed by different lowercase letters (^a^–^c^) are significantly different (*p* < 0.05), *n* = 4.

## Data Availability

Not applicable.

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
