# Peer review of "The Use of Unique, Environmental Lactic Acid Bacteria Strains in the Traditional Production of Organic Cheeses from Unpasteurized Cow’s Milk"

_molecules, 2022, doi:10.3390/molecules27031097_

Round 1

Reviewer 1 Report

Dear Author

The abstract was written qualitatively, whereas it should be organized quantitatively and mention the figures and data obtained.Introduction section sounds suitable and appropriate.

Some results can be presented by figures and charts.

Lb. brevis which was used for cheese production should be examined for biogenic amins and gas production also.

Regards

Author Response

We sincerely thank you for any comments regarding the manuscript. We made the appropriate corrections that we were able to complete.

Due to the multitude of results, we decided to write a summary based on the summary, briefly discussing the general conclusions of the study. Unfortunately, the summary has word limitations. We are not able to fit all the data in the summary.

Responding to the Reviewer's suggestion to replace some of the results with graphs or figures, we reply: We decided to include the results in tables due to the large number of results. We tried to make charts but they were not legible.

Line 17. Answering the question of whether the strains were added as starters or adjunct: The LAB strains in the production of the B1 and Os2 cheeses were the starters.

Line 17-18. In the abstract, the source of the isolation of the LAB strains was added.

I would also like to thank you for your comment on the essence of testing biogenic amines and gas production. We agree with the Reviewer on the importance of carrying out the above-mentioned tests. The cheese research was part of a project that we carried out in 2020. The project topic was: 'Using environmental lactic acid bacteria to optimize the production of organic curd cheese' and included the research we presented in the manuscript. Unfortunately, at this stage of the research, we did not plan to check the content of biogenic amines and gas production in the produced cow’s cheeses. In December 2021, we submitted another project, which is a continuation of research on the use of environmental LAB in cheese production. The scope of the follow-up covers the concept of cheese safety more broadly, including a study on the possibility of biogenic amine formation. Of course, if we receive a grant, we can extend the research to gas production as well. We do not know the results of the competition yet. At the Reviewer's request, we can send a confirmation of submitting the grant application to the Ministry of Agriculture and Rural Development in Poland (confirmation in Polish). We hope that our explanation is enough to publish the article in the Molecules journal after the corrections. We made a lot of corrections to improve the manuscript.

Reviewer 2 Report

Dear authors,

The manuscript entitled “The use of unique, environmental lactic acid bacteria strains in the traditional production of organic cheeses from unpasteurized cow's milk” is an interesting topic that could be of interest for readers. However, there are some details to take into account, which are described in the attached file.

Author Response

The authors would like to thank the Reviewer for any valuable comments on the manuscript and for the time spent on reviewing the article. All comments were carefully considered and the manuscript was revised with the utmost care.

Below are the answers to the Reviewer's questions and comments.

Abstract:

Line 22. The abbreviation for CLA and LQI has been added.

Line 25-26. The ORP abbreviation has been added.

Line 27-28. Comparison of color and texture of cheeses has been added. The sentence has been corrected to “The B1 and Os2 cheeses were lighter, less yellow, had a more intense milk and creamy aroma, were softer, more moist and elastic than AW cheese.”

Line 28-31. The last sentence that summarizes the abstract according to the reviewer's comment has been corrected. The manuscript summary was also developed to emphasize the inadequate microbiological quality of the cheeses.

Introduction:

Line 44-68. The idea behind this paragraph was to explain to the reader that we are witnessing the degradation of our microbiotic systems from year to year. This degradation affects both the environment and our human microbiota, mainly the gut. Consuming highly processed food, fast food or an excessive amount of dietary supplements leads to the depletion of our microbiota. But the human microbiome is an amazing and unique unit, and it can reproduce itself. One way to support the natural microbiota of the human body is to eat foods rich in environmental strains of lactic acid bacteria that have probiotic properties. Food processing is mainly based on ready-made starters. LAB obtained from local sources (mainly fermented and traditional foods) allows you to supplement the body's microbiota with strains that we know well, and these strains are not commonly available in food products around the world.

Line 65-66. The LAB abbreviation has been added.

The reviewer wrote: “This section lacks information on previous studies or background that serve as a reference to this study”

Response. More information on previous studies of L. brevis  B1 and L. plantarum Os2 strains has been added in the Methodology section. 3.1.2. Starters lactic acid bacteria strains preparation.

Results and Discussion

Line 168-191. The manuscript was corrected by supplementing the LQI section (2.3) with comparison and citation of other studies.

Line 324-336. More color values of the test cheeses have been added in section 2.6. Measurement of color, instrumental texture evaluation and sensory evaluation.

The reviewer wrote: “Due to the presence of S. aureus, do you consider the commercialization of this product feasible and safe?”

Response. At the moment, the pasteurized milk cow cheeses presented by us are not intended for sale. The cheeses were produced under industrial conditions, i.e. in a plant producing traditional cheeses and other products made of unpasteurized milk (including cow and goat milk). The research presented in the manuscript was part of a project supporting small, ecological enterprises in Poland (grant financed by the Ministry of Agriculture and Rural Development in Poland). As the researchers and reviewers rightly noted, the cheeses were contaminated with bacteria of the genus Staphylococcus. Even though, the risk of producing staphylococcal toxins is negligible, these cheeses could absolutely not be sold. In our research, we also decided to eliminate the sensory evaluation (taste) due to the safety of the respondents. In December 2021, we submitted another project, which is a continuation of research on the use of environmental LAB in cheese production. The scope of the follow-up covers the broader concept of cheese safety, with detailed studies planned to find a way to eliminate staphylococci. At this stage of the research, we can no longer check what was the source of contamination - whether the microbiota came from humans (in the production of artisanal cheese) or whether it was the natural microflora of unpasteurized milk (possible cow’s mastitis). It is also planned to extend the safety research, including the possibility of producing biogenic amines by Lactobacillus strains. We do not know the results of the competition yet. At the Reviewer's request, we can send a confirmation of submitting the grant application to the Ministry of Agriculture and Rural Development in Poland (confirmation in Polish).

Materials and Methods

Line 411-418. Thank you for your due consideration. The section on whey was added later and we overlooked the information on the methodology. The total number of microorganisms is mesophilic microbial count and was determined by the plate method using nutrient agar. We have completed section 3.1.1. Acid whey.

Line 420. The extension of the LAB abbreviation has been added and the information has been supplemented with citations about the strains L. brevis B1 and L. plantarum Os2.

Line 443-445. The wording of the sentence has been changed. “The first (AW treatment) was inoculated with acid whey, while the second and third were acidified with the addition of L. brevis B1 (B1 treatment) or L. plantarum Os2 (Os2 treatment), respectively.”

Line 452. The exact amount of rennet used has been added - 0.625 g of microbial rennet.

Line 454. The exact amount of water used was added - 100 mL of boiling water.

Conclusions

Line 652-658. The manuscript summary was modified to reflect the research hypotheses.

Round 2

Reviewer 2 Report

Dear Authors,

The manuscript was properly corrected.